# Spread of Carbapenem-Resistant Gram-Negatives and *Candida auris* during the COVID-19 Pandemic in Critically Ill Patients: One Step Back in Antimicrobial Stewardship?

**DOI:** 10.3390/microorganisms9010095

**Published:** 2021-01-03

**Authors:** Laura Magnasco, Malgorzata Mikulska, Daniele Roberto Giacobbe, Lucia Taramasso, Antonio Vena, Chiara Dentone, Silvia Dettori, Stefania Tutino, Laura Labate, Vincenzo Di Pilato, Francesca Crea, Erika Coppo, Giulia Codda, Chiara Robba, Lorenzo Ball, Nicolo’ Patroniti, Anna Marchese, Paolo Pelosi, Matteo Bassetti

**Affiliations:** 1Infectious Diseases Unit, Ospedale Policlinico San Martino—IRCCS per l’Oncologia, 16132 Genoa, Italy; m.mikulska@unige.it (M.M.); daniele.roberto.giacobbe@gmail.com (D.R.G.); taramasso.lucia@gmail.com (L.T.); anton.vena@gmail.com (A.V.); chiara.dentone@hsanmartino.it (C.D.); matteo.bassetti@hsanmartino.it (M.B.); 2Department of Health’s Sciences, University of Genoa, 16132 Genoa, Italy; silvidetto@gmail.com (S.D.); stef.tutino91@gmai.com (S.T.); lauretta-zau@tiscali.it (L.L.); 3Department of Surgical Sciences and Integrated Diagnostics (DISC), University of Genoa, 16132 Genoa, Italy; vincenzo.dipilato@unige.it (V.D.P.); erika.coppo@unige.it (E.C.); giuliacodda@gmail.com (G.C.); anna.marchese@unige.it (A.M.); 4Microbiology Unit, Ospedale Policlinico San Martino—IRCCS, 16132 Genoa, Italy; francesca.crea@hsanmartino.it; 5Anesthesia and Critical Care, San Martino Policlinico Hospital, IRCCS for Oncology and Neurosciences, 16132 Genoa, Italy; kiarobba@gmail.com (C.R.); lorenzo.loryball@gmail.com (L.B.); npatroniti@gmail.com (N.P.); ppelosi@hotmail.com (P.P.)

**Keywords:** antimicrobial resistance, intensive care, carbapenem-resistance, *Candida auris*, *Klebsiella pneumoniae*, *Pseudomonas aeruginosa*

## Abstract

The possible negative impact of severe adult respiratory distress caused by severe acute respiratory syndrome coronavirus-2 (SARS-CoV-2) infection (COVID-19) on antimicrobial stewardship and infection control has been postulated, but few real-life data are available. The aim of this study was to report our experience with colonization/infection of carbapenem-resistant *Pseudomonas aeruginosa* (CRPA), carbapenem-resistant *Klebsiella pneumoniae* (CR-Kp) and *Candida auris* among critically ill COVID-19 patients admitted to the intensive care unit (ICU). All COVID-19 patients admitted to the ICUs at San Martino Policlinico Hospital–IRCCS in Genoa, Italy, were screened from 28 February to 31 May 2020. One-hundred and eighteen patients admitted to COVID-19 ICUs were included in the study. Among them, 12 (10.2%) became colonized/infected with CRPA, 6 (5.1%) with *C. auris* and 2 (1.6%) with CR-Kp. All patients with CRPA received prior treatment with meropenem, and in 11 (91.7%) infection was not preceded by colonization. Four patients (66.7%) developed *C. auris* candidemia. A significant spread of resistant pathogens was observed among critically ill COVID-19 patients. Dedicated strategies are warranted to prevent horizontal spread and maintain effective antimicrobial stewardship programs in the setting of COVID-19 care.

## 1. Introduction

Since its first description in Wuhan Province at the end of 2019 [1], severe acute respiratory syndrome coronavirus-2 (SARS-CoV-2) infection has scaled to an unprecedented pandemic. Up to a third of patients with symptomatic coronavirus disease 2019 (COVID-19) may need admission to intensive care units (ICUs) because of severe respiratory failure, requiring mechanical invasive ventilation [2,3]. Open space systems have been suggested as a way of cohorting patients with COVID-19 in case of a high number of patients admitted to ICUs in order to improve direct patient care, reduce the number of personal protective equipment (PPE) used and the risk of contamination of healthcare personnel during donning and doffing [4]. However, the protective measures applied to patients with COVID-19 often do not have the main aim of preventing cross-transmission of pathogens between hospitalized patients. Before the current pandemic, the emergence of antimicrobial resistance among bacteria and fungi has been one of the worldwide leading threats [5,6,7,8,9]. To minimize the impact of antimicrobial resistance, adequate antimicrobial stewardship programs have been recommended, as well as implementation of infection control measures, based on active screening for colonization with resistant pathogens and strict contact isolation precautions for carriers to avoid horizontal spread among patients [10]. Optimizing antimicrobial prescription might be challenging among COVID-19 patients, especially critical ones, since clinical severity, imaging features and laboratory parameters make it difficult to differentiate bacterial co-infection from the mere effects of SARS-CoV-2. Likewise, implementing measures to limit the spread of resistant pathogens [11] might be particularly difficult in settings like those of COVID-19 ICUs.

The aim of this study was to report our experience with selection and subsequent cross-transmission of drug-resistant pathogens between COVID-19 patients admitted to the ICU.

## 2. Materials and Methods

### 2.1. Study Setting and Population

San Martino Policlinico Hospital–IRCCS is a 1200-bed university reference hospital in Genoa, Italy, that was chosen as regional hub for the most complex cases of COVID-19 at the beginning of the pandemic. During the pandemic, three COVID-19 dedicated ICUs were organized: one with 27 beds opened throughout the study period; a second with 12 beds opened specifically to face the pandemic on 18 March 2020 and closed on 18 April 2020 when the number of COVID-19 cases slowly decreased; and a third temporary ICU with 8 beds, which was not included in the present study. Data of all COVID-19 patients admitted to the first two ICUs during the study period between 28 February and 31 May were reviewed in order to identify the presence of drug-resistant organisms. Demographic, clinical and microbiological data were retrieved from electronical charts. Data on patients admitted to the ICU in the previous trimester (1 December 2019 to 27 February 2020) were also retrieved to allow for a descriptive comparison. During that period only the bigger ICU with 27 beds was operative.

### 2.2. Center-Specific Protocols

All patients with severe COVID-19 admitted to our hospital received antibiotic treatment against bacterial pulmonary super-/co-infection with a fifth-generation cephalosporin. Before the COVID-19 pandemic, infectious disease (ID) consultation at the ICU was provided according to clinical judgement of treating clinicians, while after the start of the pandemic the ID consultation service was potentiated, with a team of designated ID specialists providing daily consultations to support ICU colleagues with microbiological diagnosis and antimicrobial prescription. San Martino Policlinico Hospital–IRCCS infection control policy requires that all patients admitted to the ICU undergo weekly rectal swabs for the detection of carbapenem-resistant Enterobacterales (CRE) colonization. After identification of the first two cases of *C. auris* infection, screening swabs of the groin, axillary and auricular area to detect asymptomatic *Candida* colonization were carried out among all the patients admitted to the same ICU.

### 2.3. Microbiology and Definitions

Drug-resistant pathogens, for the scope of this study, were defined as carbapenem-resistant *P. aeruginosa* (CRPA) strains; carbapenem-resistant *K. pneumoniae* (CR-Kp), regardless of the molecular mechanism of resistance and all strains of *C. auris*. Resistance to meropenem (minimum inhibitory concentration (MIC) ≥8 mg/L) was considered to define carbapenem resistance for both CRPA and CR-Kp [12]. Antimicrobial susceptibility to carbapenems, aminoglycosides, fluoroquinolones and anti-pseudomonal β-lactams (including ceftolozane/tazobactam) were tested for all the strains of *P. aeruginosa* isolated. Currently, no species-specific susceptibility breakpoints are available for *C. auris.* Therefore, susceptibilities were interpreted according to the tentative breakpoints proposed by the US Centers for Disease Control and Prevention [13]. All *C. auris* strains considered in our study proved resistant to amphotericin B, and further susceptibility to azoles and echinocandins was tested. Apart from screening swabs, microbiological samples were collected upon clinical suspicion of infection. Microorganism cultured from blood, respiratory or urinary samples were identified using MALDI-TOF mass spectrometry (Vitek MS; bioMérieux, Craponne, France). Antimicrobial susceptibility testing was performed using the Vitek2 system (Biomerieux, Craponne, France), while antifungal testing was carried out by means of the Clinical and Laboratory Standards Institute microdilution method and the Sensititre YeastOne panel (Thermo Scientific, Waltham, MA, US). *C. auris* clonal relatedness was evaluated by means of whole genome sequencing (Qiagen DNeasy PowerLyzer PowerSoil Kit; Qiagen, Hilden, Germany). Given the rarity of primary *Candida* pneumonia, all isolates of *C. auris* in respiratory samples were considered as colonizers [14]. Microbiological eradication was defined as negativity of follow-up cultures alongside clinical and laboratory improvement, while clinical cure was defined as clinical improvement without evidence of microbiological eradication.

### 2.4. Endpoint

The primary endpoint was acquisition of colonization and development of infection with CRPA, *C. auris* and CR-Kp among COVID-19 patients admitted to the ICUs.

## 3. Results

### 3.1. Study Population

From 28 February to 31 May 118 patients were admitted to the COVID-19 ICUs in our hospital. Overall, the median ICU stay was 17 days (IQR 8–27 days), 74.6% of patients (*n* = 88) were male and median age was 71 years (IQR 65–78 years). The median age-adjusted Charlson Comorbidity Index was 3 (IQR 2–4). Crude in-ICU mortality observed during the study period was 42.4% (*n* = 50). Median ICU stay in the 27-bed ICU was 18 days (IQR 9–32 days) and 21 days (IQR 13–29 days) in the 12-bed ICU.

### 3.2. Colonization or Infection Rate with Resistant Organisms in COVID-19 ICU and Descriptive Comparison with Pre-COVID-19 Period

Overall, 14 patients were included in the present study, of whom 12 (10.2%) were colonized or infected with CRPA, 6 (5.1%) with *C. auris* and 2 (1.6%) with CR-Kp. Colonization or infection with resistant pathogens was observed mainly in the 27-bed ICU, where 92 patients were admitted, 13% (*n* = 12) were colonized or infected with CRPA, 6.5% (*n* = 6) with *C. auris* and 1.1% (*n* = 1) with CR-Kp. In the 12-bed ICU only one patient with pre-existing CR-Kp colonization was admitted. Among colonized/infected patients, median time from admission to first detection of resistant organism was 18 days (IQR 16–20 days), specifically 18 days (IQR 16–20 days) for CRPA and 38 days (IQR 26–41 days) for *C. auris*. The only newly acquired CR-Kp infection developed after 20 days of ICU stay. Demographics, comorbidities, sites of colonization/infection and outcomes of the 14 patients are outlined in Table 1. During the trimester prior to the COVID-19 pandemic, 159 patients were admitted to the 27-bed ICU, 56.6% of which were males (*n* = 90), with a median age of 66 years (IQR 52–75 years). The median ICU stay was 7 days (IQR 3–16 days) and crude in-ICU mortality was 13.2% (*n* = 21). During that period, four patients (2.5%) were colonized or infected with CRPA after a median of 35 days (IQR 28–47 days) and five patients (3.1%) with CR-Kp after a median of 4 days (IQR 1–13 days). No episodes of *C. auris* colonization or infection were documented, but one patient who was admitted to the ICU in February 2020 developed *C. auris* colonization 3 days after being transferred from the ICU to the surgery ward. Three of four patients (75%) with CRPA and three of five patients (60%) with CR-Kp developed infection without prior colonization.

### 3.3. Carbapenem-Resistant P. aeruginosa

The first case of CRPA infection (a lung abscess) was diagnosed on 3 April, and nine out of 11 subsequent infections developed within one month (Table 1). Almost all patients (11/12, 91.7%) had CRPA infection, while one patient, colonized at the site of previous vascular access, died shortly after isolation without developing CRPA infection. All patients with CRPA received prior meropenem therapy, with a median duration of 5 days (range 4–22 days) and within a median of 9 days from admission to the ICU (IQR 5–11 days). This consisted of an empirical treatment of a suspected bacterial superinfection in 75% of cases (*n* = 9), while the remaining three patients (25%) received meropenem for the treatment of a ventilator-associated pneumonia due to *P. aeruginosa*, and two episodes of bloodstream infection due to, respectively, *P. aeruginosa* and extended-spectrum β-lactamases producing *E. coli*. At baseline level, aminoglycoside resistance was detected in two patients (16.7%), fluoroquinolone resistance in four (33.3%), ceftazidime resistance in five (41.7%), resistance to piperacillin/tazobactam in six (50%) and resistance to ceftolozane/tazobactam in two (16.7%). Of note, four patients (33.3%) presented selective resistance to carbapenems, with retained susceptibility to all other antimicrobials. These patients had been treated for a median of 8 days (IQR 7–12) with meropenem. Moreover, 33% (three out of nine) of the patients who received ceftolozane/tazobactam as treatment, alone or as part of combination therapy, for a median of 21 days (IQR 15–42 days) developed resistance to this drug during or after treatment. Crude mortality among CRPA patients was 41.7% (*n* = 5) after a median of 42 days (IQR 25–51 days) from first detection of infection. Attributable mortality, according to caring clinicians’ judgement, was estimated to be 16.7% (*n* = 2).

### 3.4. C. auris

The first patient with *C. auris* respiratory colonization died after 8 days of ICU stay and prior to complete culture results. The subsequent two patients were diagnosed with *C. auris* respiratory tract colonization 28 and 33 days later. The screening of patients concomitantly admitted to the ICU identified three more patients colonized with *C. auris* within the following month. All the patients colonized/infected with *C. auris*, except for the first patient, received long courses of broad-spectrum antibiotics for bacterial infections before being diagnosed with *C. auris* colonization. Overall, four patients were diagnosed with *C. auris* candidemia: one with no prior colonization and three of six colonized patients (after 4, 5 and 44 days from colonization onset). All patients colonized/infected with *C. auris*, except for the first patient, were concomitantly colonized/infected with CRPA. All strains of *C. auris* identified proved to be resistant to amphotericin-B and azoles but susceptible to echinocandins. Among patients with candidemia, we report a 50% (*n* = 2) mortality after 25 days from first *C. auris* isolation. Whole genome sequencing of the strains isolated demonstrated close genetic relatedness of the strains, consistent with nosocomial transmission of the pathogen.

### 3.5. Carbapenem-Resistant K. pneumoniae

Among the two patients colonized with CR-Kp, one was a transplant recipient patient with a known CR-Kp colonization who developed ventilator-associated pneumonia caused by this colonizing strain. The other patient with CR-Kp, who had received previous empirical meropenem therapy for 20 days, developed multimicrobial ventilator-associated pneumonia with both CR-Kp and CRPA. Both CR-Kp isolates showed a multi-drug-resistant phenotype (i.e., resistant to at least three classes of antimicrobials). Observed mortality was 50% (*n* = 1) after 26 days from development of CR-Kp infection, although this was not primarily attributable to this pathogen.

## 4. Discussion

In this paper we present real-life data on high rate of colonization/infection with carbapenem-resistant *P. aeruginosa* (CRPA) and a nosocomial outbreak of *C. auris* occurring in one of the COVID-19 dedicated ICUs at our hospital. A low rate of carbapenem-resistant *K. pneumoniae* isolation was noted.

In our setting, CRPA had the most negative impact, with 10.2% of patients admitted to two COVID-19 ICUs acquiring CRPA and almost all of them developing invasive infection. Although all the cases occurred within a short period of time, it was not possible to document horizontal cross-transmission due to lack of CRPA clonality data and the impossibility of ruling out the role of selective pressure of prior treatment with meropenem. On the contrary, given the epidemiology and clonal relatedness of *C. auris* strains in our hospital, the hypothesis of nosocomial outbreak is plausible.

We speculated on possible reasons for the observation of an outbreak and a possible cluster of carbapenem-resistant pathogens. We observed a longer median ICU stay when compared to the trimester prior to the pandemic (17 vs. 7 days) and higher in-ICU crude mortality (42.4% vs. 13.2%), suggesting the clinical complexity of these patients, more prone to acquisition of nosocomial infections. Moreover, we postulated a negative impact of COVID-19 on antimicrobial stewardship programs, an issue reported in a speculative manner in various reviews and editorials. To the best of our knowledge, only one real-life report describes the acquisition of CR-Kp colonization among COVID-19 patients [15], while two centers reported an increase in *C. auris* infections in COVID-19 patients [16,17]. We supposed that the main drivers of an increase in antimicrobial resistance during COVID-19 in our hospital were the use of broad-spectrum antimicrobials and horizontal spread of resistant strains. Wide prescription of broad-spectrum antibiotics might be attributed to the difficulty of differentiating among pulmonary bacterial co-infection and viral infection alone in febrile patients with radiological evidence of mixed consolidative and interstitial patterns [18,19,20]. Moreover, the severity of clinical conditions in a setting with a high rate of extended-spectrum beta-lactamases producing bacteria might contribute to the extensive use of carbapenems for empirical therapy [21]. In addition, contact isolation measures might be suboptimal due to several reasons. First, in COVID-19 ICUs, the use of open spaces has been preferred to permit an easier management of a high number of patients [4], increasing the potential for horizontal transmission. Second, in COVID-19 units, donning with PPE is mostly focused on the protection of healthcare personnel from viral infection, with less attention to avoiding pathogen transmission between patients, who were usually not considered at risk for harboring resistant bacteria, since most of the SARS-CoV-2 infections were community-acquired. Third, despite good clinical practice recommendations, this uncomfortable PPE might be seldom renewed when caring for different patients in open spaces to limit the risk for personnel contamination during undressing and, in some cases, due to PPE shortage. Fourth, the persistence of resistant pathogens in the environment of the ICU, despite applied contact precaution measures, could not be excluded.

Despite the increase in resistant pathogens, crude mortality among critically ill patients in our ICUs was lower than that observed in other Italian centers [22] or in large international cohorts of COVID-19 patients admitted to ICUs during the early period of the pandemic [23,24].

The main limitations of this study are its retrospective design and the lack of clonality data for CRPA and CR-Kp. Additionally, a descriptive comparison with pre-COVID-19 data might be biased given various clinical profiles of patients admitted to the ICU prior to the pandemic, compared to the more homogenous population of COVID-19 ICU patients who needed longer ICU care.

## 5. Conclusions

Our report is one of the first to provide real-life data on possible selection and transmission of drug-resistant organisms among critically ill COVID-19 patients. It highlights the need for further studies to determine the precise impact of COVID-19 on antimicrobial stewardship and infection control and to develop adequate strategies to prevent further spread of antimicrobial resistance.

## Figures and Tables

**Table 1 microorganisms-09-00095-t001:** Demographics, comorbidities, sites of drug-resistant organisms’ colonization/infection and outcomes of the 14 patients described in the study.

Patient ID	Sex	Age	Comorbidities	Pathogen Isolated	Date of First Isolation	Time from ICU Admission to First Isolation	Antimicrobial Susceptibility Profile	Site of Isolation	Subsequent Infection, Type	Time from First Isolation to Infection, Days	Treatment, Drug and Duration	Microbiological Outcome at the End of Study Period	Outcome at the Last Follow-Up (Days After the First Isolation)
1	M	70	Type 2 diabetes mellitus, obesity	*C. auris*	19 March	8	-	BAL	No	N/A	N/A	N/A	Death (same day)
2	M	52	Hypertension	CRPA	3 April	19	AMG S FQ S CAZ S P/T S C/T S	BAL	Yes, lung abscess	0	C/T, 42 days	Clinical cure	Alive, discharged from ICU (26 days)
3	M	51	Hypertension	CRPA	9 April	15	AMG R FQ R CAZ R P/T R C/T R	BAL	Yes, VAP	0	HD meropenem + i.v. colistin + i.v. fosfomycin, 14 days	Clinical cure	Alive, discharged from ICU (12 days)
4	F	57	Hyperthyroidism	CRPA	9 April	16	AMG S FQ S CAZ S P/T R C/T S	Sputum	Yes, VAP	0	C/T, 16 days	Clinical cure	Death (60 days)
5	M	67	Type 2 diabetes mellitus, COPD	CRPA	12 April	16	AMG S FQ S CAZ S P/T S C/T S	BAL	Yes, BSI	2	C/T, 14 days	Microbiological eradication	Alive, discharged from ICU (16 days)
6	M	62	None	CRPA	17 April	19	AMG S FQ S CAZ S P/T R C/T S	Pleural fluid	Yes, pleural empyema and lung abscess	0	C/T + i.v. fosfomycin + HD amikacin + nebulized colistin, 42 days	Persistence of infection	Alive, still admitted to ICU (>160 days from CRPA)
				*C. auris*	13 May	45	-	Surveillance swab	Yes, BSI	44	Caspofungin, 48 days	Microbiological eradication	
7	M	69	CAD	CRPA	18 April	16	AMG S FQ S CAZ S P/T S C/T S	BAL	Yes, lung abscess	0	C/T + nebulized colistin, 48 days; then cefiderocol 3 days	Persistence of infection	Death (51 days from CRPA, 26 from *C. auris*) *
				*C. auris*	13 May	41	-	Surveillance swab	Yes, BSI	4	Caspofungin, 19 days; then amphotericin B 7 days	Microbiological eradication	
8	M	50	None	CRPA	20 April	15	AMG S FQ S CAZ R P/T R C/T S	BAL	Yes, VAP	0	C/T, 21 days	Clinical cure	Alive, discharged from ICU (24 days from CRPA; 1 day from *C. auris*)
				*C. auris*	13 May	38	-	Surveillance swab	No	N/A	N/A	N/A	
9^§^	M	66	OLT, epilepsy	CR-Kp	1 April	0	-	Surveillance swab	Yes, VAP	1	CAZ/AVI + i.v. fosfomycin, 15 days	Clinical cure	Alive, discharged from ICU (86 days)
10	M	62	Hypertension	CRPA	22 April	30	AMG S FQ S CAZ S P/T S C/T S	BAL	Yes, VAP	0	C/T, 15 days	Clinical cure	Alive, discharged from ICU (40 days from CRPA; 46 from *C. auris*)
				*C. auris*	16 April	26	-	BAL	Yes, BSI	5	Caspofungin, 24 days	Microbiological eradication	
11	M	64	Hypertension, asthma	CRPA	27 April	10	AMG S FQ S CAZ S P/T R C/T S	BAL	Yes, BSI and lung abscess	0	C/T + amikacin + i.v. fosfomycin, 28 days	Persistence of infection	Death (42 days from CRPA; 48 days from *C. auris*)
				*C. auris*	21 April	4	-	Blood culture	Yes, BSI	0	Caspofungin, 17 days; then other 12 days		
12	M	65	Hypertension, OSAS	CRPA	3 May	23	AMG S FQ R CAZ R P/T R C/T S	Blood culture	Yes, BSI	0	C/T + gentamycin, 14 days	Microbiological eradication	Alive, discharged from ICU (36 days)
13	M	63	CVI	CRPA	18 May	19	AMG R FQ R CAZ R P/T S C/T R	BAL	Yes, VAP	0	CAZ/AVI + i.v. colistin + nebulized colistin, 15 days	Persistence of infection	Death (25 days from CRPA, 26 days from CR-Kp) *
				CR-Kp	19 May	20	-	BAL	Yes, VAP	0			
14	F	67	CAD, hypertension, hypothyroidism	CRPA	21 May	55	AMG S FQ R CAZ R P/T S C/T S	Wound swab	No	N/A	N/A	N/A	Death (2 days)

AMG, aminoglycoside; FQ, fluoroquinolones; CAZ, ceftazidime; P/T, piperacillin/tazobactam; C/T, ceftolozane/tazobactam; CAZ/AVI, ceftazidime/avibactam; HD, high dose; COPD, chronic obstructive pulmonary disease; OSAS, obstructive sleep apnea syndrome; CAD, coronary artery disease; OLT: orthotopic liver transplant; CVI, common variable immunodeficiency; CRPA, carbapenem-resistant *P. aeruginosa*; BAL, bronchoalveolar lavage; VAP, ventilator-associated pneumonia; BSI, bloodstream infection; ICU, intensive care unit; N/A: not applicable. § Patient only admitted to the 12-bed ICU; * death possibly related to CRPA infection.

## Data Availability

The data presented in this study are available on request from the corresponding author. The data are not publicly available due to privacy issues.

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
