# Peer review of "Spread of Carbapenem-Resistant Gram-Negatives and Candida auris during the COVID-19 Pandemic in Critically Ill Patients: One Step Back in Antimicrobial Stewardship?"

_microorganisms, 2021, doi:10.3390/microorganisms9010095_

Round 1
Reviewer 1 Report
Discussion, beginning:
I would like to see CRPA explained here. Also, why it is not said that CR-Kp has been also tested in this study?
Line 140 - maybe addition to "baseline" needed: "at baseline level"?
Grammar:
line 49 - "form" - change to "from"
Reviewer 2 Report
Antibiotic resistance is a growing global menace. It has been intensified and complicated by the fight against COVID-19. In this manuscript, Magnasco and coworkers report the colonization/infection of the drug-resistance pathogen between COVID-19 patients admitted to the ICU. This work will be of broad interest to readers of Microorganisms journal. However, some items need to be addressed before publication.
- The primary concern is Table 1, which outlines the data of 14 patients. However, only 12 patients were colonized or infected with CRPA, C.auris, or CR-Kp in this study (Line 115). The authors must clarify the discrepancy. Besides, it would be nice if authors can mark which patient is from the 12-bed ICU in the table.
- P. aeruginosa and C.auris are not italicized in the result section and should be correct.
- Line 265-271: Conflicts of Interest is incomplete.
